# Myxoid Glomus Tumors Showing CD34 Expression: A Series of Eight Cases

**DOI:** 10.3390/diagnostics15222852

**Published:** 2025-11-11

**Authors:** Joana Sorino, Mario Della Mura, Anna Colagrande, Costantino Ricci, Giuseppe Ingravallo, Francesco Fanelli, Francesco Fortarezza, Alessio Giubellino, Gerardo Cazzato

**Affiliations:** 1Pathology Unit, Department of Precision and Regenerative Medicine and Ionian Area (DiMePRe-J), University of Bari “Aldo Moro”, Piazza Giulio Cesare 11, 70124 Bari, Italy; j.sorino@studenti.uniba.it (J.S.); mariodellamura1@gmail.com (M.D.M.); annacolagrande@gmail.com (A.C.); giuseppe.ingravallo@uniba.it (G.I.); francescofanelli92@gmail.com (F.F.); 2DIAP—Dipartimento InterAziendale di Anatomia Patologica, Maggiore Hospital AUSL Bologna, via dell’Ospedale 8, 40131 Bologna, Italy; costanricci@gmail.com; 3General Pathology and Cytopathology Unit, Department of Medicine—DMED, University Hospital of Padova, 35100 Padova, Italy; francesco.fortarezza@aopd.veneto.it; 4Department of Laboratory Medicine and Pathology, University of Minnesota, Minneapolis, MN 55455, USA; agiubell@umn.edu

**Keywords:** glomus tumor, myxoid, mast cells, CD34, perivascular neoplasm, soft tissue tumor

## Abstract

**Background:** Myxoid glomus tumors (mGTs) are an uncommon histologic pattern of glomus tumors, characterized by prominent myxoid stromal changes that may mimic a wide range of soft tissue neoplasms. Recent reports of unexpected CD34 expression in some cases have further complicated their differential diagnosis. **Objectives:** This study aimed to characterize the histopathological, immunohistochemical, and clinical features of cutaneous mGTs, with particular emphasis on CD34 expression. **Methods:** We analyzed 8 histologically confirmed cases of cutaneous mGTs underwent to a comprehensive evaluation of morphological features and immunophenotypic profile, with available clinical data. The immunohistochemical panel included smooth muscle actin (SMA), CD34, and S100. Mast cell density was assessed by tryptase in 3 cases. As controls, 8 glomus tumors without myxoid features were also examined for CD34 expression. **Results:** The cohort consisted of 8 patients (2 males, 6 females; age range 23–71 years). All tumors were located on the distal phalanges of the digits and showed extensive myxoid stromal changes. Immunohistochemistry demonstrated SMA positivity and CD34 expression in all mGTs. In contrast, none of the control GTs without myxoid stroma expressed CD34. Mast cells were consistently identified in the tested cases, predominantly within the myxoid matrix, suggesting a possible role in stromal remodeling. **Conclusions:** mGTs represent a rare but distinct histological pattern within the glomus tumor spectrum; frequent CD34 expression and mast cell infiltration appear to be characteristic features, although their biological significance remains uncertain. Recognition of these findings is essential to avoid misdiagnosis with other CD34-positive perivascular neoplasms or myxoid soft tissue sarcomas.

## 1. Introduction

Glomus tumors (GTs) are rare soft tissue neoplasms (<2%), composed of modified smooth muscle cells from the glomus body, a thermoregulatory arteriovenous structure [1]. They are most common in young adults and typically benign, though rare malignant forms exist [2]. Sporadic GTs often harbor *NOTCH* gene rearrangements, with some showing *BRAF* or *KRAS* mutations. Rare familial cases, typically presenting with multiple lesions in an autosomal dominant hereditary pattern, are associated with loss-of-function mutations in the *GLMN* gene (chromosome 1p22.1). These mutations impair the function of glomulin in vascular smooth muscle cells, leading to abnormal vascular development and tumor formation [3,4]. GTs arise mostly in the skin and superficial soft tissues of distal extremities—especially subungual, hand, wrist, and foot—but have been reported in sites like the stomach, genitourinary tract, mediastinum, bone, and lung [1,5,6]. Cutaneous lesions are typically <1 cm, reddish-blue, and painful, often triggered by cold or touch. Deep or visceral GTs usually present as nonspecific masses [3,7].

Histologically, the vast majority of GTs are benign and consist of small, uniform glomus cells with round to oval monomorphic nuclei, lightly eosinophilic cytoplasm, and well-defined borders. The solid (conventional) type is the most common, accounting for approximately 75% of cases, and is characterized by nests of glomus cells surrounding capillaries. Other histological patterns may occasionally be observed, including oncocytic, hyalinized or epithelioid forms, as well as symplastic GT, which is defined by the presence of scattered, large, hyperchromatic nuclei in the absence of mitotic activity; these nuclear changes should be interpreted as degenerative (“ancient”) alterations without prognostic significance [2,3]. Rarely, intermediate or malignant lesions are encountered, defined by the variable presence of deep location, large size, nuclear atypia, increased mitotic activity, or atypical mitoses. In this regard, Folpe et al. [8] proposed that a GT should be classified as malignant when one of the following criteria is met: deep location, size ≥ 2 cm, atypical mitoses, or marked nuclear atypia with increased mitotic activity (≥5/50 HPF). GTs of uncertain malignant potential are defined by deep location or large size (≥2 cm), or superficial location with increased mitotic activity (≥5/50 HPF). Immunohistochemically, GTs are consistently positive for vimentin (100%), SMA (99%), MSA (95%), and calponin (80%), with collagen IV and laminin expression in ~91% of cases and h-caldesmon in ~87% [9]. In contrast, markers such as S100, CD31, cytokeratin, and EMA are negative. CD34 is usually negative, although focal to diffuse positivity may be observed in some cases.

Among the different GT patterns described in the literature, myxoid GTs represent a variant uncommonly reported, although it is not clear if it represents an entity or only a spectrum of the same lesion. These lesions are considered benign and their morphological significance and the underlying mechanisms responsible for their myxoid features remain poorly understood. Furthermore, mGTs can mimic a range of benign and malignant soft tissue neoplasms with prominent myxoid stroma. Recent literature has also highlighted the unexpected expression of CD34 in some of these myxoid cases, a feature not typical of conventional GTs. This observation has led to hypotheses regarding a transitional phenotype between smooth muscle and endothelium as the possible cell of origin [10,11], and underscores the need for careful morphologic and immunohistochemical evaluation. Herein, we present a case series of 8 mGTs and analyze their clinicopathologic features in the context of previously reported literature.

## 2. Materials and Methods

This retrospective case–control study includes eight cases of cutaneous mGTs diagnosed between 2019 and 2025 at the University of Bari “Aldo Moro”, Bari, Italy, and the University of Minnesota, Minneapolis, USA, matched with other randomly selected eight cases of typical GT to compare CD34 immuno-expression. The study was conducted in accordance with the principles of the Declaration of Helsinki, and ethical approval was waived by the Institutional Review Board (IRB) due to the use of anonymized archival data and specimens (approval code 22007, approval date 5 September 2025).

All cases included in this study were histologically confirmed cutaneous GTs with available formalin-fixed paraffin-embedded (FFPE) tissue and immunohistochemical analysis was performed. All specimens were obtained from surgical excision, which also represented the definitive treatment in all patients. Cases with incomplete clinical data or poor tissue preservation were excluded. No comorbidities specifically related to glomus tumors were identified. Patient age, sex, and tumor site were retrieved from medical records and are summarized in Table 1.

Cases were identified and retrieved through pathology databases using the terms ‘glomus tumor’ and/or ‘myxoid’, and all extracted cases were re-evaluated by two dermatopathologists (G.C. and A.G.) according to the inclusion criteria, consisting of: (1) extensive myxoid stromal changes defined as the sum of ≥30% of myxoid phenomena in the stroma, or in form of multiple scattered foci always ≥30%; (2) availability of immunohistochemical staining; (3) presence of clinical data. The original immunohistochemical panel included CD34 (QBEND/10, Dako-Agilent, Glostrup, Denmark, 1:100), S-100 (Polyclonal, Dako, 1:500), Desmin (DE-R-11, Leica, Nussloch, Germany, ready to use), and SMA (ASM-1/1A4, Thermo Fisher, Waltham, MA, USA, 1:500).

Three cases of mGTs were also tested for tryptase (Leica Biosystem, clone 10D11) to identify the presence of mast cells expressed as the mean number of positive cells/10 HPF (400×).

Finally, a comprehensive literature review was also performed using PubMed, Scopus, and Web of Science databases with the keyword “myxoid glomus tumor”. All articles describing superficial/cutaneous cases were selected for discussion, in order to contextualize our findings.

## 3. Results

A total of 8 patients (2 males, 6 females; age range: 23–71 years) met the inclusion criteria (Table 1). All tumors were located on the distal phalanges of the digits, near the ungual plate. Histologically, they were characterized by nests and sheets of uniform, round glomus cells with centrally located nuclei and eosinophilic cytoplasm, embedded within an abundant myxoid stroma. A florid, delicate vascular network was also present. According to the criteria proposed by Folpe et al. [8], all cases in our series were consistent with benign lesions, lacking any features of malignancy: all tumors were superficially located, measured between 1 and 1.5 cm in diameter, showed 0–1 mitoses per 50 high-power fields (HPF), and exhibited no nuclear atypia or necrosis. Immunohistochemistry showed SMA and CD34 positivity in all cases, with CD34 staining ranging from focal to diffuse. None of the cases expressed S100 and Desmin. In contrast, all 8 cases of GT without myxoid component were totally negative for CD34.

Figure 1 and Figure 2 show some examples of mGTs with histological and immunohistochemical insights.

Moreover, tryptase showed the presence of mast cells in all three tested cases. In particular, mast cells were found within the myxoid matrix, especially at the periphery of the lesions, with counts ranging from 12 to 20 positive cells per HPF (mean count: 16) in the hotspot areas. Figure 3 shows the twoases immunostained with anti-tryptase antibody.

## 4. Discussion

GTs, first described by Masson in 1924 [7], are rare, mostly benign perivascular neoplasms arising from modified smooth muscle cells of the glomus body [2,3]. Although their smooth muscle origin is widely accepted [12,13,14,15], as confirmed by SMA diffuse positivity, the histogenesis of GTs remains incompletely understood [10,12,13,14,15,16,17,18,19,20,21]. However, occasional CD34 expression in GTs—typically considered a progenitor and endothelial marker—has been reported [22], although its functional significance remains unclear. In this regard, some authors have proposed a transitional phenotype between smooth muscle and endothelial cells, based on the co-expression of myoid markers (SMA) and vascular antigens (CD31, CD34) in glomus cells located within sinusoidal vessel walls [10,15,20,21]. Preliminary studies have demonstrated CD34 positivity in both benign and malignant GTs, particularly within the endothelial components [16,22,23,24,25,26,27,28,29], but its biological role as well as diagnostic value are still debated.

Among the histologic variants described in the literature, the myxoid subtype has emerged as a peculiar form (Table 2). While generally regarded as benign, the morphological significance and underlying mechanisms contributing to the myxoid stromal features are not fully elucidated. Furthermore, the absence of clear classificatory criteria may have led the authors not to describe this feature, also leading us not to include all cases of GTs with myxoid features published in the literature.

Tsuneyoshi and Enjoji [20] identified a myxoid subtype in 23 of 63 GTs, characterized by myxoid stroma and cords of epithelioid tumor cells arranged around vascular spaces and the frequent presence of mast cells within the stroma. They also proposed that GTs may represent a mesodermal developmental anomaly rather than a true neoplasm, based on their ultrastructural features and slow-growing clinical course. In another mGT case, Pabuççuoğlu and Lebe [25] observed CD34 and MMP-9 expression, suggesting that MMP-9, produced by both tumor cells and mast cells, may contribute to the development of the myxoid stroma through extracellular matrix remodeling. Notably, the presence of mast cells in GTs has only previously been documented in another non-cutaneous deep-seated case, also showing myxoid changes, in relation to pain generation: indeed, the presence of mast cells may play a significant algogenic role by releasing histamine and substance P [31].

Mentzel et al. [16] further demonstrated the coexpression of SMA and CD34 in 6 mGTs, underlining that this immunophenotype could complicate the differential diagnosis with other CD34-positive tumors, such as myopericytoma or solitary fibrous tumor (SFT), particularly in the presence of myxoid stroma.

Our series included eight cutaneous GTs with prominent myxoid stromal changes (≥30%), all demonstrating CD34 expression in addition to the classical immunophenotype. CD34 expression has previously been reported in mGTs, although it is not regarded as a consistent immunophenotypic feature [1,16,25]. As summarized in Table 2, several mGTs have been described in the literature, but CD34 immunostaining was not consistently performed by all authors. With this background, our intent was to demonstrate—through a relatively larger case series—that CD34 expression can represent a recurrent finding in mGTs. Therefore, our results provide additional evidence supporting the presence of CD34 immunoreactivity in this rare variant.

Histologically, mGTs are characterized by small, uniform round cells arranged in nests or sheets around capillary-sized vessels, embedded in a pale, myxoid stroma. This morphology may mimic several benign and malignant soft tissue tumors, such as superficial acral fibromyxoma, myopericytoma, angioleiomyoma, myxoid SFT, low-grade fibromyxoid sarcoma, and myxoid liposarcoma (Table 3). The presence of prominent myxoid matrix, coupled with CD34 immunopositivity, could increase the risk of misdiagnosis, both in cutaneous and deep-seated lesions, where the clinical suspicion for GTs is lower.

Superficial acral fibromyxoma, often found in acral sites like mGTs, is CD34 and CD99 positive, lacking the perivascular round cell morphology typical of GTs [32]. Angioleiomyomas with myxoid stromal changes can also mimic mGTs; however, they are composed of interlacing bundles of smooth muscle cells and demonstrate diffuse immunoreactivity for SMA and calponin, with strong and consistent positivity for h-caldesmon [33]. Desmin staining is typically variable, ranging from focal expression to complete absence in a subset of cases [33]. These tumors are characteristically negative for melanocytic markers such as HMB45 and Melan A [3]. Myopericytoma also shares a perivascular growth pattern but is predominantly characterized by concentric arrangements of spindle cells. By immunohistochemistry, it expresses SMA and h-caldesmon, while desmin and/or CD34 are only focally positive [3]. Myxoid solitary fibrous tumors, rarely cutaneous in location, have been described as CD34 and STAT6 positive, with a patternless architecture that lacks muscle marker expression [34]. Low-grade fibromyxoid sarcoma, composed of deceptively bland spindle cells within a fibrous to myxoid background, shows positivity for MUC4, a highly sensitive and specific marker [35]. Myxoid liposarcoma may mimic mGTs due to its delicate capillary vasculature and pale stroma, but it is distinguished by the presence of lipoblasts and characteristic *FUS-DDIT3* translocation [36].

Taken together, our findings support the notion that mGTs represent anhistologic pattern within the GT spectrum, characterized by unusual morphologic (myxoid matrix) and immunohistochemial (CD34-positivity) features, that may lead to diagnostic confusion. Whether they also represent a true biological distinct entity, as suggested by CD34 expression and mast cell colonization, remains an open question. Recognizing the benign behavior of these lesions is essential to prevent overtreatment and inappropriate classification. Awareness of this rare variant, as well as careful morphologic evaluation in conjunction with a targeted immunohistochemical panel, remains the cornerstone of accurate diagnosis.

## 5. Conclusions

In conclusion, our series highlights the importance of recognizing mGTs as a rare but distinct histological variant of glomus tumors, characterized by frequent CD34 expression on immunohistochemistry. These findings expand the morphologic and immunophenotypic spectrum of GTs and underline the need for the cautious interpretation of CD34 positivity in the setting of mesenchymal tumors with myxoid stroma. Accurate diagnosis relies on integrating histologic features, immunoprofiles, and clinical context, in order to avoid misclassification with more aggressive neoplasms and ensure appropriate patient management.

Future studies on mGTs should aim to clarify whether this variant represents a distinct biological entity or merely a morphologic spectrum within the glomus tumor family. Larger multicenter cohorts, supported by molecular profiling, could help elucidate the pathogenetic mechanisms underlying the myxoid stroma and the unexpected CD34 expression. In particular, the potential role of mast cells in extracellular matrix remodeling and in symptom generation deserves further investigation. Comparative genomic studies between conventional GTs and mGTs may also uncover genetic or epigenetic differences that could explain their peculiar phenotype.

## Figures and Tables

**Figure 1 diagnostics-15-02852-f001:**
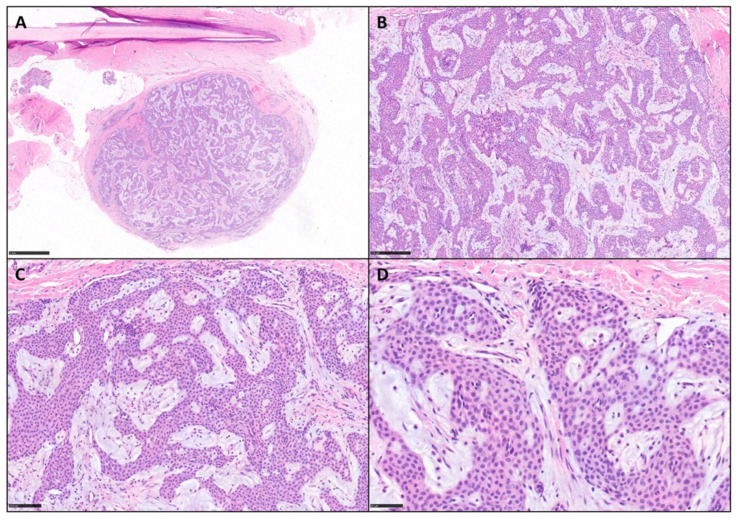
Rarely, some glomus tumors harbor abundant myxoid changes. (**A**) Photomicrograph showing an example of peri-ungueal mGT (H&E, scale bar 1 mm). (**B**,**C**) Tumor is made up of nests and cords of glomus cells embedded in fibrous and myxoid stroma (H&E, scale bar 250 and 100 μm). (**D**) Higher magnification of small, uniform, round glomus cells with centrally located nuclei, amphophilic to lightly eosinophilic cytoplasm, and sharply defined borders immersed in a myxoid stroma (H&E, scale bar 50 μm).

**Figure 2 diagnostics-15-02852-f002:**
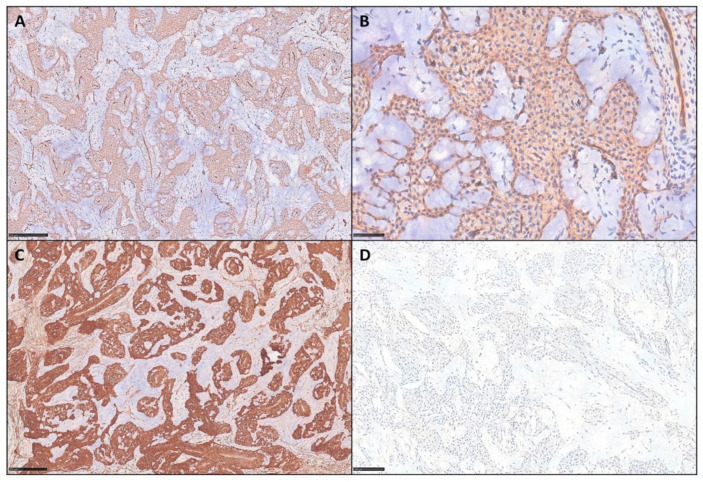
(**A**,**B**) Tumor cells, in these two representative cases of mGTs from our series, show diffuse cytoplasmic expression of CD34 with moderate intensity (IHC for CD34, scale bar 250 μm and 50 μm). (**C**) All cases displayed strong and diffuse positivity for SMA (IHC for SMA, scale bar 250 μm), (**D**) as well as negativity for Desmin (IHC for Desmin, scale bar 100 μm).

**Figure 3 diagnostics-15-02852-f003:**
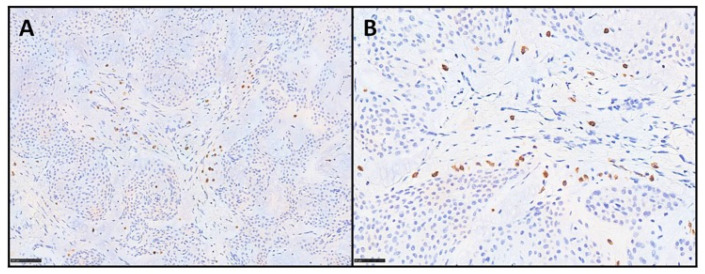
(**A**,**B**) Mast cells are highlighted by tryptase immunostaining in the tested cases. Positive cells are mainly distributed within the myxoid stroma, as single cells or small loose clusters (IHC for tryptase, scale bar 100 and 50 μm).

**Table 1 diagnostics-15-02852-t001:** Clinicopathological and immunohistochemical features of mGTs.

Case Number	Age	Sex	Site	IHC	Mastocytes Count (Tryptase)
Case 1	33	M	IV left toe, distal phalanx	SMA+, CD34+ (focal), S100−	20/HPF
Case 2	55	F	I left toe, distal phalax	SMA+, CD34+, S100−	15/HPF
Case 3	26	F	I left finger, distal phalanx	SMA+, CD34+, S100−	12/HPF
Case 4	53	F	III right finger	SMA+, CD34+, S100−	NP
Case 5	71	M	II left finger, nail bed	SMA+, CD34+, S100−	NP
Case 6	32	F	III left finger, nail bed	SMA+, CD34+, S100−	NP
Case 7	23	F	II right finger, distal phalanx	SMA+, CD34+, S100−	NP
Case 8	69	F	IV left finger, nail bed	SMA+, CD34+, S100−	NP

Abbreviations: M, male; F, female; SMA, smooth muscle actin; HPF, high-power field; NP, not performed. Note: Mast cell count was evaluated by tryptase immunostaining and expressed as the mean number of positive cells/10 HPF (×400).

**Table 2 diagnostics-15-02852-t002:** Summary of published cases of mGTs.

Authors (Year of Publication)	No. of Cases	Location	Histologic Description	Immunohistochemistry
Tsuneyoshi and Enjoji (1982) [20]	23	Fingers	Glomus epithelioid cells around vessels within myxoid matrix and numerous mast cells	NR
Pabuççuoğlu & Lebe (2008) [25]	1	Finger	Typical glomus cells in myxoid stroma containing mast cells	SMA+, CD34+, MMP-9+
Mentzel et al. (2002) [16]	6	Fingers	Solid, glomangioma and glomangiomyoma types with prominent myxoid change	SMA+, CD34+
Da Silva et al. (2018) [30]	1	Finger	Glomus cells with symplastic features within myxoid stroma	SMA+
Our cases (2025)	8	Fingers	Typical glomus cells in myxoid stroma containing numerous mast cells	SMA+, CD34+, tryptase+, S100−

Summary of the reported cutaneous mGTs cases in the literature, listed in chronological order, detailing number of cases, anatomic location, histologic features, and immunohistochemical findings. CD34 expression is frequently observed in the myxoid pattern, as well as mast cells colonization within the myxoid stroma. NR: not reported.

**Table 3 diagnostics-15-02852-t003:** Immunohistochemical profile of mGTs and their main mimickers.

Entity	CD34	SMA	Desmin	S100	Other Markers
**Myxoid** **glomus tumor**	Variable/focal to diffuse positivity	Positive	Negative	Negative	MSA+, vimentin+
**Superficial acral fibromyxoma**	Diffuse positivity	Negative or focal	Negative	Negative	EMA+
**Angioleiomyoma**	Negative	Positive	Positive	Negative	Calponin+, SMA+, HMB45/Melan-A−
**Myopericytoma**	Rarely positive or negative	Positive	Positive	Negative	h-caldesmon+
**Solitary fibrous tumor (myxoid variant)**	Diffuse positivity	Negative	Negative	Negative	STAT6+
**Low-grade fibromyxoid sarcoma**	Negative	Negative	Negative	Negative	MUC4+
**Myxoid liposarcoma**	Negative	Negative	Negative	Negative	MDM2+

The table outlines the main immunohistochemical patterns of mGTs compared with their most common histologic mimics. Overall, mGTs display a distinctive profile characterized by consistent smooth muscle marker expression and frequent CD34 positivity, an unusual but recurrent feature in this variant. In contrast, the other entities show different combinations of markers, often lacking SMA or expressing lineage-specific antigens such as STAT6, MUC4, or MDM2, which facilitate their recognition. In general, the table highlights the crucial role of an integrated immunohistochemical approach in guiding the differential diagnosis of myxoid lesions.

## Data Availability

The original contributions presented in this study are included in the article. Further inquiries can be directed to the corresponding author.

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
