# Peer review of "Myxoid Glomus Tumors Showing CD34 Expression: A Series of Eight Cases"

_diagnostics, 2025, doi:10.3390/diagnostics15222852_

Round 1
Reviewer 1 Report
Comments and Suggestions for Authors
The authors have investigated CD34 expression in rare glomus tumors. The study included eight observations, which is both a limitation and an important consideration for clinical oncology due to the extremely rare nature of this disease.
The authors describe glomus tumors in the introduction but do not distinguish their subtypes—solid, angiomatous, and myxoid. They only mention the solid subtype. Furthermore, in the introduction, the authors describe criteria for the malignancy of glomus tumors, although their study did not assess the number of mitoses or tumor nodule size. This information may be redundant.
The materials and methods section is poorly written. There is no information on the study's compliance with the Declaration of Helsinki or on approval by the ethics committee.
There are no clinical characteristics of the patients, including treatment received, comorbidities, and inclusion and exclusion criteria. The histological and immunohistochemistry examinations are poorly described. Was the material obtained by biopsy or surgical resection?
The table titles are usually located at the top. Table 1 contains a typo (NP - not reported). The table title should be concise. A legend should be placed below the table. How were mast cells counted?
The figures show low-magnification micrographs. Figure 1 requires higher-magnification micrographs to clearly show the tumor cell morphology.
The CD34 micrographs should be presented at higher magnification. CD34 expression in this figure is questionable. It would also be better to show the mast cells at higher magnification.
The list of references contains only 3 sources published from 2021 to 2025.
Author Response
Response to reviewer 1
We sincerely thank the reviewer for the valuable comments and constructive suggestions, which have helped us to improve the clarity and scientific quality of the manuscript. We have revised the text accordingly, as detailed below.
- Glomus tumor subtypes (Introduction)
Reviewer comment:
The authors describe glomus tumors in the introduction but do not distinguish their subtypes—solid, angiomatous, and myxoid. They only mention the solid subtype.
Response:
We thank the reviewer for this observation. In the last WHO Classification of Skin Tumors 5th Edition, solid glomus tumours is recognized as “entity” while it’s not clear if the oncocytic, epithelioid, myxoid and hyalinized forms are really histological variants or only patterns. For this reason we have decided to underscore the solid type, with also the addition of this concept. Thank you.
- Criteria for malignancy
Reviewer comment:
The authors describe criteria for the malignancy of glomus tumors, although their study did not assess the number of mitoses or tumor nodule size. This information may be redundant.
Response:
The reference to the malignancy criteria was included primarily to provide a brief overview of glomus tumor classification and to clarify that all tumors in our series fulfilled the histopathological criteria for benign glomus tumors. To prevent any possible misunderstanding, we have further clarified this point in the Results section, specifying that the Folpe classification was used solely to confirm the benign nature of our cases.
Modified text (Results):
According to the criteria proposed by Folpe et al. [8], all cases in our series were consistent with benign lesions, lacking any features of malignancy: all tumors were superficially located, measured between 1 and 1.5 cm in diameter, showed 0–1 mitoses per 50 high-power fields (HPF), and exhibited no nuclear atypia or necrosis.
- Materials and Methods
Reviewer comment:
The materials and methods section is poorly written. There is no information on the study's compliance with the Declaration of Helsinki or on approval by the ethics committee.
Response:
Ethical compliance and institutional approval have been added.
Added text (Materials and Methods):
“This retrospective case-control study includes eight cases of cutaneous mGTs diagnosed between 2019 and 2025 at the University of Bari “Aldo Moro”, Bari, Italy, and the University of Minnesota, Minneapolis, USA, matched with other randomly selected eight cases of typical GT to compare CD34 immuno-expression. The study was conducted in accordance with the principles of the Declaration of Helsinki and ethical approval was waived by the Institutional Review Board (IRB) due to the use of anonymized archival data and specimens”.
- Clinical characteristics, inclusion criteria and specimen source
Reviewer comment:
There are no clinical characteristics of the patients, including treatment received, comorbidities, and inclusion and exclusion criteria.
Was the material obtained by biopsy or surgical resection?
Response:
Clinical and methodological information was already included in the Materials and Methods section, but we agree that it required clearer presentation. We have therefore revised the text to explicitly state the inclusion criteria, source of specimens, and treatment details.
Added text (Materials and Methods):
“All cases included in this study were histologically confirmed cutaneous GTs with available formalin-fixed paraffin-embedded (FFPE) tissue and performed immunohistochemical analysis. All specimens were obtained from surgical excision, which also represented the definitive treatment in all patients. Cases with incomplete clinical data or poor tissue preservation were excluded. No comorbidities specifically related to glomus tumors were identified. Patient age, sex, and tumor site were retrieved from medical records and are summarized in Table 1.”
- Table formatting and mast cell quantification
Reviewer comment:
The table titles are usually located at the top. Table 1 contains a typo (NP - not reported). The table title should be concise. A legend should be placed below the table. How were mast cells counted?
Response:
We appreciate these observations. We have corrected the table formatting (title above, legend below), fixed the typo, and added a description of mast cell counting (see material and methods, Table 1, Table 2 and Table 3).
- Figure quality and magnification
Reviewer comment:
The figures show low-magnification micrographs. Figure 1 requires higher-magnification micrographs to clearly show the tumor cell morphology.
Response: We have provided a new panel of pictures with higher magnification that show tumor cell morphology.
- CD34 micrographs
Reviewer comment:
The CD34 micrographs should be presented at higher magnification. CD34 expression in this figure is questionable. It would also be better to show the mast cells at higher magnification.
Response: We have provided new CD34 pictures and also mast-cells immunohistochemistry.
- References
Reviewer comment:
The list of references contains only 3 sources published from 2021 to 2025.
Response: Thank you for this observation. We have added the citation to the most recent 5th Edition of WHO Classification of Skin Tumors; hovewer, being a rare histological pattern of mGTs, there are no new references in the last years
Reviewer 2 Report
Comments and Suggestions for Authors
Reviewer Comments:
The manuscript titled “Myxoid Glomus Tumors Showing CD34 Expression: A Series of Eight Cases” presents an interesting and relevant topic for both pathologists and clinicians. The study is well-conceived and clearly written. However, a revision is recommended to enhance the presentation and clarity of the data.
-
Figure Quality:
- Figure 1A–C appears out of focus and lacks optimal resolution. I recommend the authors replace Figure 1 with higher-quality images showing representative features of myxoid glomus tumor at magnifications of 40×, 100×, and 200×.
- Figure 2A–C shows poor immunohistochemical staining quality, particularly with high background in panel C. A revised Figure 2 is recommended, including improved staining for SMA and at least one additional smooth muscle marker. One figure demonstrating CD34 expression is sufficient.
-
CD34 Expression:
- CD34 expression in glomus tumors has been previously described and is not considered a typical finding (see PMID: 18788860). This should be acknowledged and discussed in the manuscript.
-
Differential Diagnosis:
- The differential diagnosis should include dermal nevus and paraganglioma. While low-grade fibromyxoid sarcoma and myxoid liposarcoma do not closely resemble glomus tumors histologically, their mention in the differential is acceptable.
-
Conclusion (Line 248):
- Please revise the sentence:
“In conclusion, our case series highlights the importance of recognizing mGTs as a rare but distinct variant of GTs, characterized by frequent CD34 expression”
to:
“In conclusion, our case series highlights the importance of recognizing mGTs as a rare but distinct variant of GTs, characterized by frequent CD34 expression by immunohistochemistry.”
- Please revise the sentence:
Acceptable
Author Response
Response to reviewer 2
- Figure Quality
Reviewer comment:
Figure 1A–C appears out of focus and lacks optimal resolution. I recommend the authors replace Figure 1 with higher-quality images showing representative features of myxoid glomus tumor at magnifications of 40x, 100x, and 200x.
Response: We have provided a new panel of pictures with higher magnification that show tumor cell morphology.
Reviewer comment:
Figure 2A–C shows poor immunohistochemical staining quality, particularly with high background in panel C. A revised Figure 2 is recommended, including improved staining for SMA and at least one additional smooth muscle marker. One figure demonstrating CD34 expression is sufficient.
Response: We have provided new pictures of immunohistochemistry. We don’t include other smooth muscle stainings as they were negative (as we have specified in the main text).
- CD34 Expression
Reviewer comment:
CD34 expression in glomus tumors has been previously described and is not considered a typical finding (see PMID: 18788860). This should be acknowledged and discussed in the manuscript.
Response:
Our intent was precisely to demonstrate, through a larger case series, that CD34 expression is a consistent feature in myxoid glomus tumors. As summarized in Table 2, although several myxoid glomus tumors have been previously described in the literature, not all authors performed CD34 immunostaining in their analyses.
In fact, even in the reference suggested (PMID: 18788860) - which was already cited in our manuscript - the authors refer to the work of Mentzel et al. (6 cases) for CD34 expression data. When these are combined with the 1 additional CD34-positive case reported by Pabuççuoğlu & Lebe (also listed in Table 2), the total amounts to only 7 CD34-positive myxoid glomus tumors described to date. We expanded the paragraph within the Discussion where we had already provided information about CD34 expression.
Added text (Discussion):
“Our series includes eight cutaneous GTs with prominent myxoid stromal changes (≥30%), all demonstrating CD34 expression in addition to the classical immunophenotype. CD34 expression has previously been reported in mGTs, although it is not regarded as a consistent immunophenotypic feature [1,16, 25]. As summarized in Table 2, several mGTs have been described in the literature, but CD34 immunostaining was not consistently performed by all authors. With this background, our intent was to demonstrate - through a relatively larger case series - that CD34 expression can represent a recurrent finding in mGTs. Therefore, our results provide additional evidence supporting the presence of CD34 immunoreactivity in this rare variant.”
- Differential Diagnosis
Reviewer comment:
The differential diagnosis should include dermal nevus and paraganglioma. While low-grade fibromyxoid sarcoma and myxoid liposarcoma do not closely resemble glomus tumors histologically, their mention in the differential is acceptable.
Response:
Dear reviewer, thank you very much for your suggestion, but we prefer to not include these entities because we focused only on mGT differential diagnosis, so we included the neoplasms with myxoid aspects. Thank you.
- Conclusion (Line 248)
Reviewer comment:
Please revise the sentence:
“In conclusion, our case series highlights the importance of recognizing mGTs as a rare but distinct variant of GTs, characterized by frequent CD34 expression”
to:
“In conclusion, our case series highlights the importance of recognizing mGTs as a rare but distinct variant of GTs, characterized by frequent CD34 expression by immunohistochemistry.”
Response:
The sentence in the Conclusion has been revised as following: “In conclusion, our series highlights the importance of recognizing mGTs as a rare but distinct histological variant of glomus tumors, characterized by frequent CD34 expression on immunohistochemistry”.
Round 2
Reviewer 1 Report
Comments and Suggestions for Authors
The authors have significantly improved the article; the manuscript can be accepted for publication.